# Preparation of Dihydromyricetin-Loaded Self-Emulsifying Drug Delivery System and Its Anti-Alcoholism Effect

**DOI:** 10.3390/pharmaceutics15092296

**Published:** 2023-09-08

**Authors:** Jianxia Dong, Shu Wang, Jiamin Mao, Zhidan Wang, Shiying Zhao, Qiao Ren, Jialing Kang, Jing Ye, Xiaohong Xu, Yujin Zhu, Quan Zhang

**Affiliations:** 1Department of Medicinal Natural Products, West China School of Pharmacy, Sichuan University, Chengdu 610041, China; ghdsacfds@gmail.com (J.D.); wang_shu@scu.edu.cn (S.W.); 2Department of Pharmacy, West China Hospital, Sichuan University, Chengdu 610041, China; wangzhidan@scu.edu.cn (Z.W.); jialingkang1992@gmail.com (J.K.); 3Institute of Materia Medica, Structure-Specific Small Molecule Drugs Key Laboratory of Sichuan Provincial Universities, School of Pharmacy, Chengdu Medical College, Chengdu 610500, China; zhaoshiying326@gmail.com (S.Z.); renqiao98@gmail.com (Q.R.); cy2259569759@gmail.com (J.Y.); xuxiaohong@cmc.edu.cn (X.X.); yujinzhu9@gmail.com (Y.Z.); 4Development and Regeneration Key Lab of Sichuan Province, Department of Pathology, Department of Anatomy and Histology and Embryology, Chengdu Medical College, Chengdu 610500, China; 5Chengdu Nature’s Grace Biological Technology Co., Ltd., Chengdu 610213, China

**Keywords:** dihydromyricetin, self-emulsifying drug delivery system, oral bioavailability, lymphatic absorption, anti-alcoholism

## Abstract

Intraperitoneal injection of dihydromyricetin (DMY) has shown promising potential in the treatment of alcoholism. However, its therapeutic effect is limited due to its low solubility, poor stability, and high gut-liver first-pass metabolism, resulting in very low oral bioavailability. In this study, we developed a DMY-loaded self-emulsifying drug delivery system (DMY-SEDDS) to enhance the oral bioavailability and anti-alcoholism effect of DMY. DMY-SEDDS improved the oral absorption of DMY by facilitating lymphatic transport. The area under the concentration-time curve (AUC) of DMY in the DMY-SEDDS group was 4.13-fold higher than in the DMY suspension group. Furthermore, treatment with DMY-SEDDS significantly enhanced the activities of alcohol dehydrogenase (ADH) and acetaldehyde dehydrogenase (ALDH) in the liver of mice (*p* < 0.05). Interestingly, DMY-SEDDS also increased ADH activity in the stomach of mice with alcoholism (*p* < 0.01), thereby enhancing ethanol metabolism in the gastrointestinal tract and reducing ethanol absorption into the bloodstream. As a result, the blood alcohol concentration of mice with alcoholism was significantly decreased after DMY-SEDDS treatment (*p* < 0.01). In the acute alcoholism mice model, compared to saline treatment, DMY-SEDDS prolonged the onset of LORR (loss of righting reflex) (*p* < 0.05) and significantly shortened the duration of LORR (*p* < 0.01). Additionally, DMY-SEDDS treatment significantly reduced gastric injury in acute alcoholism mice. Collectively, these findings demonstrate the potential of DMY-SEDDS as a treatment in the treatment of alcoholism.

## 1. Introduction

Alcohol, classified as a class A carcinogen by the World Health Organization (WHO), poses a global problem due to alcohol abuse. Annually, alcohol consumption claims the lives of approximately 3 million people worldwide, accounting for 5.3% of all deaths. It ranks as the third most significant public health concern after cancer and cardiovascular diseases [1].

Ethanol, being lipophilic, readily traverses cell membranes, facilitating its rapid absorption in the digestive tract (30% in the stomach and 70% in the small intestine) following ingestion. Absorbed alcohol circulates through the bloodstream, reaching various tissues and organs. When ethanol enters the gastrointestinal tract, a portion undergoes first-pass metabolism (FPM) through alcohol dehydrogenase (ADH) in gastric mucosa [2,3]. Unmetabolized ethanol subsequently enters the bloodstream and reaches the liver via the inferior vena cava. The majority of ethanol metabolism occurs in the liver, primarily by ADH and acetaldehyde dehydrogenase (ALDH), leading to its decomposition into water and carbon dioxide [4]. ADH and ALDH, predominantly found in the liver and gastric mucosa, play an important role in ethanol metabolism [5]. As a result, the activities of ADH and ALDH directly influence the concentration of ethanol and acetaldehyde in the body.

Excessive alcohol consumption can lead to alcoholism, causing damage to tissue and organs in the digestive, nervous, cardiovascular, and other systems [6]. Among these, damage to the digestive system is particularly common. Gastric mucosal injury is clinically observed in patients with alcoholism, presenting as stomach discomfort, nausea, vomiting, and, in severe cases, hematemesis and gastric perforation [7]. On one hand, the high alcohol concentration, owing to its lipid-soluble properties, directly harms the deep mucosa beneath the upper cortex. It damages the mucus barrier and disrupts the physiological environment, leading to gastric congestion, bleeding, erosion, ulceration, destruction of the lamina propria, and even tissue necrosis. On the other hand, alcohol stimulates the release of reactive oxygen species (ROS) from the gastric mucosa, which disturbs oxidative stress balance and impairs superoxide dismutase (SOD) function [8]. Consequently, the excessive ROS accumulates, exacerbating tissue damage. Simultaneously, malondialdehyde (MDA), as a final lipid peroxidation product, affects nuclear factor NF-kB, inducing increased expression of inflammatory factors and alterations in cell membrane fluidity and permeability. These changes further compromise tissue structure and function [9]. During inflammation and subsequent healing processes following injury [10], alcohol intensifies oxidative stress response through increased endotoxin leakage and proinflammatory cytokine release from immune and non-immune cells. Overall, alcohol directly influences cellular physiological functions and indirectly affects systemic oxidative and inflammatory responses through alcohol metabolism [11,12]. Hence, scavenging a wide range of ROS in the body may aid in slowing the progression of alcohol-induced oxidative damage [13]. Clearly, the key to combating alcoholism lies in reducing alcohol absorption, accelerating its metabolic rate, increasing SOD levels, and removing lipid peroxides to mitigate the damage inflicted by alcohol and its metabolites on bodily organs, thereby preserving overall health.

Dihydromyricetin (DMY) is a dihydroflavonol compound extracted from the grapevine genus, with Rattan tea being a particularly rich source [14]. Recent reports have demonstrated that DMY possesses diverse pharmacological effects, including anti-inflammatory [15], antioxidant [16], anti-bacterial [17], anti-tumor [18], blood lipid-regulating [19], and liver-protective properties [20]. Studies have indicated that intraperitoneal injection of DMY can effectively combat alcoholism and withdrawal symptoms in rats [21]. Despite its excellent anti-alcoholism activity and high safety profile [22,23], the polyphenolic structure of DMY significantly reduces its stability, water solubility (0.2 mg/mL) [24], and short half-life [25]. Furthermore, our preliminary experiments have revealed that P-glycoprotein (P-gp) efflux [26] and gastrointestinal first-pass effect [27] lead to the low bioavailability of oral DMY. These limitations greatly hinder the clinical application of DMY for anti-alcoholism and other therapeutic purposes. Therefore, selecting an appropriate drug delivery system to mitigate P-gp efflux and the gastrointestinal first-pass effect while increasing oral bioavailability holds significant importance for maximizing the efficacy of DMY.

A self-emulsifying drug delivery system (SEDDS) represents an isotropic mixture comprising lipids, surfactants, cosurfactants, and drugs [28]. Upon contact with aqueous media such as gastric and intestinal fluids, SEDDS spontaneously forms micro-nano emulsions under gentle agitation [29]. Compared to liposomes and nanoparticles, SEDDS contains higher surfactant content, which can modulate gastrointestinal permeability to enhance oral drug absorption [30]. Additionally, as the drug delivery system lacks an aqueous phase, its formulation exhibits relatively stable characteristics. Certain surfactants present in SEDDS can also improve oral absorption by inhibiting P-gp activity [31]. Moreover, within the SEDDS, the drug can be absorbed through the lymphatic system, bypassing the liver’s first-pass effect. Consequently, SEDDS holds great promise in addressing the oral absorption challenges of DMY and augmenting its anti-alcoholism efficacy.

## 2. Materials and Methods

### 2.1. Materials

DMY (purity > 98%) was purchased from Shanghai Yuanye Bio-technology Co., Ltd. (Shanghai, China). Isopropyl myristate, polyoxyethylene hydrogenated castor oil, and diethylene glycol monoethyl ether were provided by Shanghai Aladdin Biochemical Technology Co., Ltd. (Shanghai, China). Cycloheximide (CYC) was obtained from Chengdu Huaxia Chemical Reagent Co., Ltd. (Chengdu, China). Total superoxide dismutase (T-SOD) and malondialdehyde (MDA) kits were provided by Nanjing Jiancheng Bioengineering Institute (Nanjing, China). All other chemicals were of analytical or high-performance liquid chromatography (HPLC) grade.

### 2.2. Animals

Sprague-Dawley (SD) rats (male, 180–220 g) and male Kunming mice (18–22 g) were provided by Chengdu Dashuo Laboratory Animal Co., Ltd. (Chengdu, China). All animals were exposed to natural light for 12 h under environmental conditions of 25 ± 2 °C and humidity at 75 ± 5%. Prior to the experiments, the animals were fasted for 12 h and had access to water ad libitum.

### 2.3. Analysis of DMY by HPLC-MS/MS

The concentration of DMY was analyzed using a high-performance liquid chromatography/mass spectrometer triple quadrupole system (HPLC-MS/MS, Agilent Technologies, Santa Clara, CA, USA). Nitrogen was used as the drying gas with a temperature of 350 °C and a flow rate of 11 L/min. The triple-quadrupole mass spectrometer was operated in positive electrospray ion source (ESI) mode, and multiple reaction monitoring (MRM) was selected. The transition of m/z for DMY was 319.0→192.9. The optimal parameters of the MS/MS were as follows: capillary voltage, 4 kV; atomizing gas pressure, 15 psi; cracking voltage, 100 V; collision energy, 5 V; collision pool acceleration voltage, 7 V.

Chromatographic separation was performed using an Agilent XDB-C18 column (4.6 × 50 mm, 1.8 μm). The mobile phase consisted of 22% acetonitrile and 78% distilled water containing 0.1% formic acid. The mobile phase was delivered at a flow rate of 0.3 mL/min and a column temperature of 30 °C. The injection volume was 2 μL.

### 2.4. Analysis of DMY by HPLC

A Shimadzu HPLC system (LC-2030, Shimadzu, Kyoto, Japan) equipped with an ultraviolet detector was used to detect DMY. Chromatographic separation was performed on the ZORBAX SB-C18 column (150 mm × 4.6 mm, 5 μm) with the mobile phase delivered at a flow rate of 1.0 mL/min. The initial mobile phase consisted of 74% A (water containing 0.1% phosphate) and 26% B (methanol). The gradient elution procedure was as follows: from 26% to 98% B in 0–15 min; from 98% to 26% B in 15–40 min; at 40–50 min, 26% B. The temperatures of the autosampler and column were set at 15 °C and 30 °C, respectively. The detection wavelength was 292 nm. The injection volume was 10 μL.

### 2.5. Analysis of Alcohol by GC

The concentration of alcohol in the blood was determined using a flame ionization detector (FID) of an Agilent gas chromatograph (GC, Agilent Technologies, Santa Clara, CA, USA) equipped with an automated headspace injection sampler. Separation was performed on an Agilent 19091J-413: HP-5 (30 m × 320 μm × 0.25 μm) capillary column. Nitrogen (99.99%) was used as the carrier gas with a constant flow rate of 12 mL/min. The temperature of the headspace oven was set to 85 °C and equilibrated for 20 min. The sample was injected in split mode (split ratio: 20:1) within 0.5 min. The inlet sample temperature and detector temperature were set at 200 °C and 220 °C, respectively. The heating program was as follows: the initial temperature was held at 40 °C for 5 min, then heated at a rate of 26.4 °C/min up to 250 °C.

### 2.6. Preparation of DMY-SEDDS

DMY-SEDDS was prepared using the dissolution method. The excipients (see Appendix A and Figure A1), namely isopropyl myristate, polyoxyethylene hydrogenated castor oil, and diethylene glycol monoethyl ether, were mixed in a ratio of 25:40:35 (*w*/*w*/*w*). Each excipient was weighed according to this ratio and mixed thoroughly by swirling. Then, 20 mg of DMY was added for every 1 g of excipients, and the mixture was vortexed until DMY was completely dissolved. The flowchart for preparing DMY-SEDDS is shown in Figure 1.

### 2.7. Physicochemical Characterization of DMY-SEDDS

The particle size and Zeta potential of DMY-SEDDS were determined using the Malvern System (MalvernZS-90, Worcestershire, UK). The following steps were followed: 1 g of DMY-SEDDS was gently stirred and heated at 37 °C, then diluted with 100 mL of distilled water and transferred to the sample chamber. The temperature was set at 25 °C, and the measurements were repeated three times. The results were expressed as the average value.

The morphology of DMY-SEDDS was observed and photographed using TEM (Hitachi, Japan). Prior to analysis, DMY-SEDDS was diluted 100-fold with distilled water at 37 °C and spotted onto a copper mesh. It was allowed to air dry, and the copper mesh was placed on a wax plate. Then, it was negatively stained with 2% phosphotungstic acid for 15 min, washed with distilled water, and gently blotted dry with filter paper.

### 2.8. Determination of EE% and DL%

The EE% and DL% of DMY-SEDDS were measured using the ultrafiltration centrifugation method [32]. DMY-SEDDS was dissolved in distilled water to form an oil-in-water emulsion. The samples were divided into three parts for parallel analysis. 0.5 mL of the emulsion was added to a 3000 Dalton ultrafiltration centrifuge tube provided by Labgic Technology Co., Ltd. (Hefei, China) and centrifuged at 3552× *g* for 10 min. After centrifugation, the lower layer of the centrifuge tube was diluted ten times with methanol. The amount of free drug was analyzed and detected by HPLC. The EE% and DL% of DMY-SEDDS were calculated as follows:(1)$$EE(\%)=\frac{Amount\;of\;DMY\;encapsulated\;in\;the\;SEDDS}{Added\;amount\;of\;DMY}\times 100$$
(2)$$DL(\%)=\frac{Amount\;of\;DMY\;encapsulated\;in\;the\;SEDDS}{Total\;amount\;of\;DMY-SEDDS}\times 100$$

### 2.9. Effect of Medium pH on the Emulsification of DMY-SEDDS

In this experiment, the stability of DMY-SEDDS emulsification was studied at the pH values of 1.3 (stomach) and 7.8 (intestine). A total of 1 mL of DMY-SEDDS (20 mg/g; each gram of SEDDS contains 20 mg of DMY) was added to 50 mL of phosphate buffer solution with the respective pH values. The emulsion was gently stirred at 50 rpm for 30 min at a water bath temperature of 37 °C. The particle size of the emulsion was determined using the Malvern System.

### 2.10. Stability Analysis in Simulative Gastrointestinal Fluid

The amount of 0.2 mL of DMY-SEDDS (20 mg/g, each gram of SEDDS contains 20 mg of DMY) and 20 mg of DMY (accurately weighed and dissolved in 1 mL of absolute ethanol) were added to 7.8 mL of SGF [33] containing 7.80 g of NaCl, 0.37 g of CaCl2, 0.35 g of KCl, 0.02 g of MgCl_2_, 0.32 g of NaH_2_PO_4_, 1.40 g of glucose, 1.37 g of NaHCO_3_, and 3.84 mg of pig pepsin (3200–4500 U/mg). Hydrochloric acid was used to adjust the pH to 1.3 (*n* = 3) [34]. The timer was started when the simulated gastric fluid was added. After shaking to emulsify, 0.1 mL of the liquid was collected at 0, 10, 30, 60, 90, 120, 180, 240, 360, 480, and 1440 min, respectively. An equal volume of hydrochloric acid solution with pH 0.6 was added to each sample. After mixing, the samples were filtered through a 0.22 μm filter membrane and analyzed by HPLC. Each sample was analyzed independently in triplicate.

SIF was prepared according to the reported method, with some modifications [35]. The SIF consisted of 7.80 g of NaCl, 0.37 g of CaCl_2_, 0.35 g of KCl, 0.02 g of MgCl_2_, 0.32 g of NaH_2_PO_4_, 1.40 g of glucose, 1.37 g of NaHCO_3_, and 19.2 mg of bile salt. The pH was adjusted to 6.8 using 0.5 mol/L NaOH or diluted hydrochloric acid. Sampling was conducted at 0, 10, 30, 60, 90, 120, 180, 240, 360, 480, and 1440 min. The specific procedure was described above.

The drug concentration at 0 min was considered 100%. The retention rate of DMY (R%) was calculated using the following formula to determine the stability of DMY-SEDDS in simulative gastrointestinal fluid.
(3)$$R(\%)=\frac{DMY\;concentration\;at\;the\;sampling\;time\;point}{DMY\;concentration\;at\;0\;min\;point}\times 100$$

### 2.11. In-Situ Single-Pass Intestinal Perfusion

The in-situ single-pass intestinal perfusion (SPIP) model accurately simulates the intestinal absorption of oral drugs [36]. The SPIP model was employed to further investigate the absorption characteristics of DMY-SEDDS, with the Ka and Papp as evaluation indexes in vivo. Eighteen SD rats were randomly divided into three groups (group 1: DMY; group 2: DMY-SEDDS; group 3: CYC + DMY-SEDDS). For group 3, one hour before the experiment, a dimethylsulfoxide solution of CYC (0.6 mg/mL) was intraperitoneally injected at a dose of 3 mg/kg. After anesthesia induced by intraperitoneal injection of 5% chloral hydrate solution (7 mL/kg), the abdominal cavity was opened along the midabdominal line to expose the intestinal segment. The jejunum was opened at both ends, cannulated, and ligated. The intestinal segment was connected to a peristaltic pump (LabF6 type, Baoding Shenchen Pump Industry Co., Ltd., Baoding, China). The intestine was flushed with normal saline at 37 °C at a flow rate of 0.2 mL/min and emptied with air. The entire system was saturated with blank Krebs-Ringer buffer at 37 °C for 30 min, and then the freshly prepared drug solution (40.0 μg/mL DMY solution and DMY-SEDDS in Krebs-Ringer buffer) was pumped at the same speed. After equilibration, the experiment was initiated and timed. The effluent was collected within 15 min of the start of the formal experiment. At the end of the experiment, the intestinal segments were excised, and the circumference (c) and length (l) of each intestinal segment were measured to calculate the radius (r) and volume (v) of the perfused intestinal segment. The collected samples were analyzed by HPLC.

Gravimetric analysis was used to correct the inflow and outflow volumes of the enema fluid in order to eliminate the effect of variations in enema fluid volume. The Ka and Papp of DMY in the rat intestinal tract were calculated using the following formula:(4)$$Ka=\left(1-\frac{Cout\left(m_{2}-m_{1}\right)}{Cin\left(m_{3}-m_{4}\right)}\right)\frac{Q}{\pi r^{2}l}$$
(5)$$Papp=\frac{-QIn\left(\frac{Cout\left(m_{2}-m_{1}\right)}{Cin\left(m_{3}-m_{4}\right)}\right)}{2\pi rl}$$

Cin and Cout represented the mass concentrations of the perfusion fluid at the inlet and outlet of the intestine, respectively, measured in μg/mL. The variables l and r denoted the length (in cm) and cross-section radius (in cm) of the irrigated intestinal segment, while Q represented the perfusion velocity (in mL/min). The values m_1_ and m_2_ indicated the mass (in grams) of the empty collection tube and the collection tube receiving liquid at the outlet after a certain duration of perfusion, respectively. Similarly, m_3_ and m_4_ represented the mass (in grams) of the tube when the inlet of intestinal flow was filled with perfusion fluid before perfusion and the mass of the remaining liquid and tube after a certain period of perfusion.

### 2.12. Pharmacokinetic Study

Eighteen SD rats were randomly assigned to three experimental groups, with six animals in each group. The groups were as follows: group 1, DMY suspension; group 2, DMY-SEDDS; and group 3, CYC + DMY-SEDDS. One hour prior to the experiment, the animals in group 3 were treated with CYC (3 mg/kg, i.p.), while the other groups received an equal volume of saline. Rats in group 1 were administered DMY suspension orally at a dose of 100 mg/kg. Group 2 and group 3 received DMY-SEDDS orally at the same dose of 100 mg/kg. Blood samples were collected from the suborbital venous plexus of the rats at 0.033, 0.083, 0.25, 0.5, 1, 1.5, 2, 4, 8, and 24 h. The samples were placed in preheparinized centrifuge tubes. After blood collection, the samples were cryocentrifuged at 3500 rpm, and 100 μL of supernatant was taken. To precipitate proteins completely, 300 μL of acetonitrile with 1% formic acid was added, followed by vortexing for 10 min. The mixture was then centrifuged at 12,000 rpm for 10 min, and 2 μL of the supernatant was injected into the HPLC-MS/MS system for analysis.

### 2.13. Study on Anti-Alcoholism of DMY-SEDDS

#### 2.13.1. Loss of Righting Reflex Assay

Thirty-six male Kunming mice were randomly divided into three groups: the model group, the DMY group, and the DMY-SEDDS group. Prior to the experiment, all groups underwent a 12-h fasting period with free access to water. The model group received oral administration of normal saline, while the other groups were orally treated with DMY suspension (100 mg/kg) and DMY-SEDDS (100 mg/kg) per mouse. After one hour, all animals were orally administered 56% alcohol (0.18 mL/10 g). The LORR onset time was calculated as the time from finishing alcohol administration to the onset of LORR. LORR ended when the animal was capable of pronating three times within a 30-s period. The duration of LORR was measured as the time from the beginning of LORR until the animal regained the ability to resume normal movement. Mouse with LORR phenomenon was judged to be drunkenness. The number of mice with drunkenness was recorded. The drunkenness rate of mice was calculated using the following formula:(6)$$Drunkenness\;rate\left(\%\right)=\frac{Number\;of\;mice\;with\;drunkenness}{Total\;number\;of\;mice\;in\;the\;experiment}\times 100$$

#### 2.13.2. Determination of Alcohol Concentration in Blood

Seventy-two male Kunming mice were randomly divided into three groups: the model group, the DMY group, and the DMY-SEDDS group. Prior to the experiment, all groups underwent a 12-h fasting period. These mice were treated with saline, DMY suspension (100 mg/kg), or DMY-SEDDS (100 mg/kg) orally, respectively. After one hour of administration, 56% alcohol (0.14 mL/10 g) was orally administered. At 5, 60, 180, and 300 min after alcohol administration, eyeballs were removed to obtain a minimum of 0.5 mL of blood. The blood samples were placed in headspace flasks and analyzed for blood ethanol concentration using a GC.

#### 2.13.3. Effects of DMY-SEDDS on ADH and ALDH Activities in Gastric and Liver 

##### Tissues in Mice

Twenty-four male Kunming mice were randomly divided into four groups: the normal group, the model group, the DMY group (100 mg/kg), and the DMY-SEDDS group (100 mg/kg). The normal group and model group received oral administration of distilled water (0.05 mL/10 g), while the other two groups received the corresponding test substance according to body weight for one week. After the final administration, all mice were orally gavaged with 56% alcohol (0.14 mL/10 g). Two hours later, the mice were euthanized by spinal dislocation. The stomach and liver were then separated, rinsed with cold normal saline, dried with filter paper, weighed, and homogenized in normal saline to obtain a 10% tissue homogenate. The supernatant was obtained through centrifugation at 3000 rpm/min at 4 °C, and the activities of ADH and ALDH in the stomach and liver were measured using appropriate assay kits.

#### 2.13.4. Protective Effect of DMY-SEDDS on Gastric Mucosal Injury Induced by Ethanol in Mice

Fifty male Kunming mice were randomly assigned to five groups. All mice fasted for 12 h without access to water. The blank control group and the model group received oral administration of normal saline (0.05 mL/10 g). The experimental groups received oral administration of DMY and DMY-SEDDS (100 mg/kg, 0.05 mL/10 g). One hour later, absolute ethanol (0.1 mL/10 g) was administered via intragastric gavage in all groups except the blank control group. One hour later, the mice were euthanized, and the stomachs were immediately dissected. The stomachs were cut along the greater curvature, washed with normal saline to remove blood clots and gastric fluid, and then spread on filter paper for evaluation of the GUI according to the Guth method [37], and then photographed. The Guth method was employed to measure the GUI in mice, providing an evaluation of the degree of gastric mucosal injury [38]. The abnormal conditions of gastric mucosa integrity, surface smoothness, color, and bleeding in each group of mice were observed and assessed based on the Guth scoring standard [39]. The severity was determined by the size and number of gastric mucosa bleeding points, with scores ranging from 0 (no pathology) to 8 (large ulcer > 6 mm). The total score of all bleeding points on the gastric mucosa of each mouse represented the GUI.

Formalin-fixed gastric tissue was sectioned, dehydrated, embedded in wax, and stained with hematoxylin and eosin. The sections were observed under a light microscope (Eikon Eclipse ci, Japan).

The remaining stomachs were rinsed with pre-cooled normal saline, blotted dry with filter paper, weighed, and homogenized in pre-cooled normal saline. The resulting 10% stomach tissue homogenate was centrifuged at 3000 rpm for 15 min at a low temperature. The supernatant was stored at −80 °C for T-SOD and MDA detection. The activity of SOD and the content of MDA in the gastric homogenate were determined using the hydroxylamine method and TBA (thiobarbituric acid) method [40]. All procedures followed the instructions provided by the Nanjing Jiancheng Bioengineering Institute (Nanjing, China). The results were adjusted for the total protein content of gastric tissue.

### 2.14. Statistic Analysis

All data were presented as average value ± standard deviation (SD). Statistical significance was analyzed by one-way analysis of variance (ANOVA) and Student’s *t*-test in the SPSS system.

## 3. Results and Discussion

### 3.1. Physicochemical Characterization of DMY-SEDDS

The mean particle size, polydispersity index (PDI), and zeta potential of the prepared DMY-SEDDS were (36.23 ± 1.44) nm, (0.127 ± 0.004), and (−3.16 ± 0.32) mV, respectively, after emulsification with water. As depicted in Figure 1A, the particle size of DMY-SEDDS emulsified with water reached the nanometer scale [41], exhibiting a highly uniform distribution. Transmission electron microscopy (TEM) analysis (Figure 1B) revealed that DMY-SEDDS formed spherical emulsion droplets with a uniform shape after dilution with water. Furthermore, the encapsulation efficiency (EE%) and drug loading (DL%) of the preparation were determined using the ultrafiltration centrifugation method. The EE% and DL% of the preparation were (92.48 ± 1.14)% and (1.84 ± 0.44)%, respectively, indicating that the drug was predominantly present in the oil-in-water structure with minimal leakage and high encapsulation efficiency.

### 3.2. Effect of Medium pH on the Emulsification of DMY-SEDDS

Table 1 presents the results of the particle size analysis of DMY-SEDDS under different pH conditions, revealing that the particle size remained consistent across various pH media. These findings suggest that the microemulsion formed upon dilution with gastric and intestinal fluids remains highly stable and unaffected by pH after oral administration.

### 3.3. Stability Analysis in Simulative Gastrointestinal Fluid

Due to its multiple ortho phenolic hydroxyl groups, DMY is known to exhibit pH instability and is prone to degradation under near-neutral conditions [42]. Figure 1C demonstrates the excellent stability of DMY solution and DMY-SEDDS in simulated gastric fluid (SGF). The DMY content in both formulations did not significantly change after one day of digestion in SGF with a pH of 1.2. However, at pH 6.8 (Figure 1D), DMY exhibited instability. The DMY content in both the DMY solution and DMY-SEDDS gradually decreased over time. Specifically, the DMY solution experienced a rapid drop in content within 60 min, with a degradation rate of 24.9%. In contrast, DMY-SEDDS showed only a 5.36% decline in DMY content. Moreover, after four hours of digestion in simulated intestinal fluid (SIF), DMY-SEDDS retained 72.36% of DMY, whereas DMY solution retained only 53.09%. This enhanced stability of DMY-SEDDS can be attributed to the oil-in-water structure formed by emulsifying DMY in water, which provides better protection against degradation compared to DMY solution. The DMY-SEDDS formulation developed in this study effectively improves the stability of DMY in the intestinal environment, thereby enhancing its oral bioavailability.

### 3.4. In-Situ Single-Pass Intestinal Perfusion

In our study, we used the chylomicron flow-blocking model to explore whether DMY-SEDDS improves bioavailability by enhancing DMY lymphatic absorption. Previous research conducted by other scientists demonstrated a strong correlation between vitamin D3 lymphatic absorption and the mesenteric lymphatic cannulation model, using two chylomicron blockers, CYC and Pluronic L-81. Remarkably, these two chylomicron blockers exhibited minimal side effects [43]. Also, Sun M et al. blocked chylomicron blood flow by intrabitoneal injection of CYC to inhibit lymphatic transport in the self-microemulsifying drug delivery system [44]. CYC, a non-specific inhibitor of protein synthesis, inhibits chylomicron secretion from epithelial cells to the chylomicron by acting on microtubules [45]. As shown in Figure 2, DMY-SEDDS demonstrated greater absorption compared to free DMY solution during intestinal perfusion, with both the absorption rate constant (Ka) and the apparent permeability (Papp) for DMY-SEDDS (Ka = 33.60, Papp = 3.30) higher than those of DMY solution (Ka = 28.82, Papp = 2.85). According to the report by Arik Dahan et al. [46], CYC was used to investigate lymphatic absorption in rats. Pharmacokinetic tests conducted in normal rats, rats with a mesenteric lymphatic cannulation model, and rats pretreated with CYC revealed that the reduction in total absorption after CYC administration corresponded to the reduction in lymphatic absorption. By comparing the jejunal absorption of DMY-SEDDS alone and DMY-SEDDS in combination with CYC, Ka, and Papp decreased by 33.28% and 25.98%, respectively. These results suggest that DMY-SEDDS enhances intestinal absorption of DMY by promoting lymphatic absorption when DMY-SEDDS and DMY solution have similar solubilities.

### 3.5. Pharmacokinetic Study

As shown in Figure 3, at each time point, the DMY concentration in blood for the DMY-SEDDS group was significantly higher than that of the DMY suspension group, indicating the effective increase in plasma DMY levels achieved by DMY-SEDDS.

The plasma drug concentrations were analyzed and calculated using DAS 3.0 software, and the main pharmacokinetic parameters are presented in Table 2. The maximum plasma concentration (Cmax) of DMY for the DMY-SEDDS group was 438.76 ± 192.46 μg/L, which was 2.08 times higher than that of the DMY suspension group (211.01 ± 114.71 μg/L). Furthermore, the area under the concentration-time curve from 0 to 24 h (AUC_(0–24 h)_) for the DMY-SEDDS group (1256.59 μg∙h/L) was 4.13 times greater than that of the DMY suspension group (304.06 μg∙h/L) (*p* < 0.01). Moreover, the mean residence time (MRT) of DMY-SEDDS (6.31 h) in the blood was 1.60 times longer than that of DMY suspension (3.94 h). After cycloheximide (CYC) injection, the Cmax of DMY-SEDDS decreased by 46.04%, and the AUC_(0–24 h)_ decreased by 38.77%. This decline can be attributed to the inhibition of DMY lymphatic absorption by CYC. These results indicate that DMY-SEDDS enhances lymphatic absorption, slows down elimination, and prolongs the systemic circulation of the drug in the body, significantly improving the oral bioavailability of DMY.

### 3.6. Study on Anti-Alcoholism of DMY-SEDDS

Drunkenness, commonly known as alcoholism, is characterized by central nervous system depression resulting from excessive alcohol consumption. Clinically, drunkenness is divided into three stages: the excitement period, the ataxia period, and the coma period [47]. In the model group, mice exhibited signs of drunkenness, such as irritability, unsteady gait, slow movement, lethargy, and syncope, after ingesting 56% ethanol. As shown in Table 3, the alcoholism rate was 100% in both the model group and the DMY group, while it was 75% in the DMY-SEDDS group. Furthermore, deaths occurred in the model group and the DMY group, but no deaths were observed in the DMY-SEDDS group. Oral administration of DMY-SEDDS significantly prolonged the onset time of loss of righting reflex (LORR) in mice compared to the model group and the DMY group (*p* < 0.05) (Figure 4A). Additionally, DMY-SEDDS significantly reduced the duration of LORR compared to DMY suspension (*p* < 0.01). These findings indicate that DMY-SEDDS exhibits a superior anti-alcoholism effect compared to DMY suspension and improves the survival rate of mice with alcoholism.

The effectiveness of DMY-SEDDS in combating alcoholism was further evaluated by measuring blood alcohol concentration in mice. As depicted in Figure 4B, the blood alcohol concentration peaked at 180 min and increased rapidly from 0 to 60 min. The rise in blood alcohol concentration was slower in mice treated with DMY-SEDDS compared to the other two groups. Starting from 180 min, the blood alcohol concentration began to decrease, and at both 180 and 300 min, the blood alcohol concentration in the DMY-SEDDS group (180 min: 5.30 mg/mL, 300 min: 2.32 mg/mL) was notably lower than that of the DMY group (180 min: 7.01 mg/mL, 300 min: 5.03 mg/mL) (*p* < 0.01) and the model group (180 min: 6.53 mg/mL, 300 min: 4.13 mg/mL) (*p* < 0.05). These results indicate that DMY-SEDDS can reduce the blood alcohol content, thereby exerting its anti-alcoholism effect.

ADH and ALDH are crucial enzymes involved in alcohol metabolism in the body, mainly present in the stomach and liver [48]. The presence of ADH isoenzymes in the gastric mucosa helps in the removal of ethanol before it is absorbed into the bloodstream, thereby reducing serum ethanol concentration [49]. Unmetabolized ethanol passes through the systemic circulation to the liver, where 90% of ethanol is metabolized, primarily by the ethanol dehydrogenase oxidation system [50]. Therefore, the activity of the ethanol dehydrogenase system determines the rate of ethanol elimination from the body. In this study, the anti-alcoholism mechanism of DMY-SEDDS was further elucidated by measuring the ADH and ALDH levels in the liver and stomach.

It has been reported that DMY can increase the expression of ethanol-metabolizing enzymes in vivo and in vitro. Specifically, the expression levels of ADH and ALDH in HepG2 cells treated with DMY. At the same time, ADH and ALDH were increased in the liver of mice treated with DMY [51]. Figure 5 shows that the activity of ADH in the stomach and liver of mice in the DMY-SEDDS group was significantly increased compared with the model group and DMY group (*p* < 0.05). Moreover, the ALDH activity in the stomach and liver tissues of the DMY-SEDDS group was increased by 34.7% and 87.4%, respectively, compared with the model group (*p* < 0.05). However, the ALDH activity in the stomach or liver tissues was not significantly different between the DMY group and the model group. These results suggest that DMY-SEDDS can enhance the ADH and ALDH activities in the gastric mucosa of mice with alcoholism, thereby accelerating the clearance of ethanol in the stomach and reducing its absorption into the bloodstream. Additionally, DMY-SEDDS promotes ethanol elimination by increasing the ADH and ALDH activities in the liver, leading to a decrease in serum ethanol concentration and reducing the toxic effects of alcohol on various organs in the body.

Clinically, gastric mucosal bleeding caused by alcoholism is a common occurrence, with patients experiencing symptoms such as nausea, vomiting, abdominal discomfort, and, in severe cases, even fatal hematemesis [52]. To simulate normal human drinking through intragastric administration, this study investigated the protective effect of DMY-SEDDS on ethanol-induced gastric mucosal injury in mice. The overall morphology of gastric tissue was observed, and the injury degree of gastric mucosal was assessed using the gastric injury index (GUI) and gastric oxidative stress factors.

Figure 6A demonstrates that the gastric mucosa of normal mice appeared intact, with normal shape, color, and abundant mucus. In contrast, the gastric epithelium of ethanol-treated mice exhibited an uneven surface, obvious congestion and edema of the gastric mucosa, and significant bleeding. The pathological results (Figure 6B) revealed severe damage to the gastric mucosa in the model group, whereas the mice in the DMY-SEDDS group showed reduced degeneration and necrosis of gastric mucosal epithelial cells and glands compared with the model mice and DMY-treated mice. As shown in Figure 6C, the GUI value in the model group was 20.6 ± 8.00, indicating that the alcohol-induced gastric injury mice model was successfully established. DMY treatment reduced gastric bleeding, erosion, and ulceration, resulting in a 30.58% reduction in alcohol-induced gastric injury (GUI = 14.3 ± 2.79). Mice treated with DMY-SEDDS exhibited significantly reduced gastric bleeding and a GUI value of 8.2 ± 5.84 (*p* < 0.01), corresponding to a 60.19% inhibition rate of gastric mucosal injury. Additionally, compared to the DMY suspension group, DMY-SEDDS significantly decreased the GUI value (*p* < 0.05), indicating the better protective effect of DMY-SEDDS against the gastric mucosal injury induced by ethanol.

Excessive alcohol consumption overwhelms the capacity of ethanol-metabolizing enzymes, leading to the induction of oxidative stress and the formation of lipid hydroperoxides. For gastric mucosa, ROS play a significant role in causing damage. Lipid hydroperoxides and MDA can damage the structure of cell membranes, leading to cell swelling and necrosis [53]. The MDA level indirectly reflects the extent of the ROS attack. SOD acts as an antioxidant enzyme, scavenging ROS generated by oxidative stress and playing a crucial role in maintaining the balance between oxidation and antioxidant mechanisms in organisms [54]. In general, the SOD value indirectly indicates the scavenging ability of oxygen-free radicals.

Figure 6D,E demonstrate that ethanol significantly reduces SOD levels in the gastric tissue and increases the gastric tissue level of MDA, indicating high oxidative stress levels induced by ethanol. Compared to the model group, the MDA level in the gastric tissue of mice treated with DMY-SEDDS was dramatically reduced (*p* < 0.01), and the SOD level was notably improved (*p* < 0.01). DMY suspension treatment also significantly reduced the level of MDA in the stomach (*p* < 0.01) while only slightly increasing the level of SOD. In contrast, DMY-SEDDS significantly increased the level of SOD in gastric tissue (*p* < 0.05) compared to the DMY suspension group. These results suggest that the therapeutic effect of DMY-SEDDS on gastric mucosal injury is closely related to the antioxidant effect of DMY.

## 4. Conclusions

In this study, SEDDS were employed to enhance the stability of DMY in the gastrointestinal tract, promote lymphatic absorption, improve oral bioavailability, and enhance efficacy against alcoholism and alcohol-induced gastric mucosal injury. The emulsion droplets formed by DMY-SEDDS, when diluted with water, exhibited spherical shapes with an average particle size of 36.23 ± 1.44 nm, a PDI of 0.127 ± 0.004, and a zeta potential of −3.16 ± 0.32 mV. DMY-SEDDS demonstrated greater gastrointestinal stability compared to DMY alone. Pharmacokinetic and in-situ single-pass intestinal tests showed that DMY-SEDDS relied on intestinal lymphatic absorption, bypassing portal vein absorption and thereby avoiding partial first-pass effects. This led to enhanced oral bioavailability of DMY and prolonged drug retention time in the body. In comparison to DMY suspension, DMY-SEDDS significantly improved the activity of ADH and ALDH in gastric tissue, facilitating the elimination of ethanol in the stomach and reducing its absorption. Furthermore, DMY-SEDDS enhanced the activity of ADH and ALDH in the liver, promoting the oxidative metabolism of ethanol and acetaldehyde and reducing the concentration of ethanol in the blood. The prolonged onset time and shortened duration of the LORR indicated an increased survival rate of mice with acute alcoholism. Moreover, DMY-SEDDS exhibited superior pharmacodynamic performance compared to DMY suspension by increasing SOD levels and decreasing MDA levels, thereby exerting an antioxidant effect and providing better protection against ethanol-induced gastric mucosal injury. Consequently, DMY-SEDDS represents a promising approach for improving the oral bioavailability of DMY and facilitating its clinical application in the treatment of alcoholism. DMY-SEDDS is expected to become a potential clinical drug for treating alcoholism and protecting against alcohol-induced gastric mucosal injury.

## Data Availability

The authors confirm that the data supporting the findings of this study is available within the article.

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
