# Peer review of "Preparation of Dihydromyricetin-Loaded Self-Emulsifying Drug Delivery System and Its Anti-Alcoholism Effect"

_pharmaceutics, 2023, doi:10.3390/pharmaceutics15092296_

Round 1

Reviewer 1 Report

In this study, author developed DMY-loaded SEDDS for the treatment of alcoholism. This seems to be a novel therapeutic approach for alcoholism, and significant advantage is obtained by preparing SEDDS. Although the draft is overall well-written, some revision should be addressed before publication as listed below.

1. In method section following points are unclear. If they are not common, some explanation or citation should be included.

Please be more specific about “3000 Dalton ultrafiltration centrifuge tube”

“8000 rpm” should be converted to g, or describe the diameter of rotor.

Please clarify the unit of “DMY-SEDDS (20 mg/g)”

Please specify or include some citations of “Guth method”.

2. How dose the author think about the release profile of DMY from SEDDS after re-dispersed.

3. in 3.3, pH 1.3 may be 1.2.

4. in 3.3, It would be easier to understand if following sentence were moved to the beginning of the paragraph. “Due to its multiple ortho phenolic hydroxyl groups, DMY is known to exhibit pH instability and is prone to degradation under near-neutral conditions”

5. in 3.4, are there any reports that show CYC inhibit absorption of SEDDS?

6. It would be easier to understand if the order of 3.4 (PK profile) and 3.5 (in situ test) were reversed.

7. in 3.6, author concluded that DMY-SEDDS enhance the ADH and ALDH activity, but does DMY really have such a distinctive function?

8. headline of “5. Patent” should be removed.

Author Response

Response to reviewers

Thank you for offering us an opportunity to improve the quanlity of our submitted manuscript (Preparation of dihydromyricetin-loaded self-emulsifying drug delivery system and its anti-alcoholism effect). We appreciated very much the reviewers’ constructive and insightful comments. In this revision, We have tried our best to improve and make some changes in the manuscript. The yellow part that has been revised according to your comments.

On the next pages, our point-to-point responses to the queries raised by the reviewers are listed.

Comment 1: In method section following points are unclear. If they are not common, some explanation or citation should be included. Please be more specific about “3000 Dalton ultrafiltration centrifuge tube”. “8000 rpm” should be converted to g, or describe the diameter of rotor. Please clarify the unit of “DMY-SEDDS (20 mg/g)”. Please specify or include some citations of “Guth method”.

Response: Thanks for your advice. We have described in detail about “3000 Dalton ultrafiltration centrifuge tube” in this revision. And 8000 rpm has been converted to 3550 g. The unit of “DMY-SEDDS (20 mg/g)” has been clarified in this revision. We added some citations about Guth method , which are cited in [38], [39].

Comment 2: How dose the author think about the release profile of DMY from SEDDS after re-dispersed.

Response: When DMY-SEDDS entered gastric fluid to spontaneously form oil-in-water nanoemulsion, the oil-in-water structure wrapped DMY in the oil phase core, preventing the release of drugs from the oil phase to gastric and intestinal fluid. Moreover, the nanoparticles formed after spontaneous emulsification of DMY-SEDDS were small enough, it was speculated that the drug was absorbed by the intestinal lymphatic system in the form of oil-in-water.

Comment 3: In 3.3, pH 1.3 may be 1.2.

Response: Thanks for pointing out the mistake. We have modified pH 1.3 to pH 1.2.

Comment 4: In 3.3, It would be easier to understand if following sentence were moved to the beginning of the paragraph. “Due to its multiple ortho phenolic hydroxyl groups, DMY is known to exhibit pH instability and is prone to degradation under near-neutral conditions”.

Response: Thanks for your suggestion. The sentence has been moved to the beginning of the paragraph in 3.3.

Comment 5: In 3.4, are there any reports that show CYC inhibit absorption of SEDDS?

Response: Yes, there is. We have already discussed this issue in page 10 of this revision.

Comment 6: It would be easier to understand if the order of 3.4 (PK profile) and 3.5 (in situ test) were reversed.

Response: Thanks for your suggestion. We reversed the order of 3.4 and 3.5 in our revision.

Comment 7: In 3.6, author concluded that DMY-SEDDS enhance the ADH and ALDH activity, but does DMY really have such a distinctive function?

Response: Yes, it dose. We have already discussed this issue in page 14 of this revision.

Comment 8: Headline of “5. Patent” should be removed.

Response: Thanks for your suggestion. We have corrected the mistake.

Reviewer 2 Report

The article “Preparation of dihydromyricetin-loaded self-emulsifying drug delivery system and its anti-alcoholism effect” is focuses on a developing and evaluating dihydromyricetin loaded self-emulsifying drug delivery system to enhance the oral bioavailability and anti-alcoholism effect. The work is novel.

The comments are as follows:

1.       Abstract: Intraperitoneal should not be bold.

2.       Reference style in text need to verify as per journal guidelines.

3.       All methods need to cite if they are not developed inhouse.

4.       2.6. Preparation of DMY-SEDDS, how authors got this formula?  No preliminary study and physicochemical characterization is three with various of ratio or oil, cosolvent and surfactant. Provide the supplementary file for better understanding.

5.       Figure 1: TEM image is not clear.

6.       Why 2.9. Effect of Medium pH on the Emulsification of DMY-SEDDS, this study is important? Is drug shows pH dependent release?

7.       Table 1: What is percentage in this and why all 100% as result?

8.       Figure 2. Mean plasma concentration-time profiles of DMY in rats after oral administration of DMY and DMY-SEDDS (each value represents the mean ± SD, n = 6). There are graphs in graphs, please make it A and B form. Its difficult to understand.

9.       Table 2: Why in AUC and Cmax deviation is so high?

10.   What is purpose of using cycloheximide injection?

11.   How Drunkenness rate determined in Table 3?

12.   Conclusions need to be specific. Please cut short to important outcomes. 

Reviewer 3 Report

The authors investigated a dihydromyricetin-loaded self-emulsifying drug delivery system (DMY-SEDDS) for the treatment of alcoholism. Among other findings, they found that ethanol metabolism was enhanced in the gastrointestinal tract and reduced ethanol absorption in the bloodstream of model mice. As a result, the blood alcohol concentration of mice with alcoholism was significantly reduced after treatment with DMY-SEDDS.

I have a few minor suggestions to improve the paper:
1.    A schematic figure of the DMY molecules and the preparation of the nanoparticles will help the reader to better understand the present Fig. 1B. Especially the question why there are large and small nanoobjects in Fig. 1B needs a better answer. Are the small objects just artifacts?
2.     With reference to the sentence in the Conclusion section, " DMY-SEDDS is expected to become a potential clinical drug for the treatment of alcoholism and protection from alcohol-induced gastric mucosal injury.": Typically, clinical drugs should have low polydispersity and high repeatability of the preparation process. Besides the size distribution in Fig. 1A, is there any other evidence that the polydispersity of DMY-SEDDS is low?
3.    During the discussion, the authors could expand the context available to readers by referring to analogous systems based on lipid nanocarriers for drug delivery: 'Cubosomal lipid nanoassemblies with pH-sensitive shells created from biopolymer complexes: A synchrotron SAXS study', Journal of Colloid and Interface Science, 2022, 607, 440-450. https://doi.org/10.1016/j.jcis.2021.08.187
